# Resilience and Regulation of Emotions in Adolescents: Serial Mediation Analysis through Self-Esteem and the Perceived Social Support

**DOI:** 10.3390/ijerph19138007

**Published:** 2022-06-29

**Authors:** Janusz Surzykiewicz, Sebastian Binyamin Skalski, Agnieszka Sołbut, Sebastian Rutkowski, Karol Konaszewski

**Affiliations:** 1Faculty of Philosophy and Education, Catholic University of Eichstaett-Ingolstadt, 85072 Eichstaett, Germany; 2Faculty of Education, Cardinal Wyszynski University in Warsaw, 01938 Warsaw, Poland; s.skalski@uksw.edu.pl; 3Faculty of Education, University of Bialystok, 15328 Bialystok, Poland; a.solbut@uwb.edu.pl (A.S.); k.konaszewski@uwb.edu.pl (K.K.); 4Education and Social Readaptation Youth Centre in Goniadz, 19110 Goniadz, Poland; sebastian.rutkowski@mowgoniadz.pl

**Keywords:** resilience, self-esteem, emotion regulation, social support, adolescents

## Abstract

The aim of this study was to test a model that takes into account self-esteem and perceived social support as potential mediators of the relationship between resilience and emotional regulation. The study involved 251 adolescents aged between 14 and 19 years (M = 16.85). The study procedure consisted of completing paper-and-pencil questionnaires to measure resilience, self-esteem, and answer questions about perceived social support and emotional regulation. Bootstrap sampling analysis showed statistically significant serial mediation (B = 0.030; *p* < 0.001). As a result of the analysis, a positive direct relationship between resilience and emotional regulation was observed (B = 0.061; *p* < 0.001). Our results suggest that self-esteem and perception of social support may mediate the relationship between resilience and emotional regulation. The findings have an applicable value. They can be used to develop preventive and educational programs, as well as therapeutic interventions. The obtained results show that interventions aimed at resilience can improve self-assessment and perceived social support and thus favor the high level of emotional regulation skills in the adolescent group.

## 1. Introduction

Growing up is an important stage in human life. Due to unbalanced physiological and psychological development, young people often struggle with problems related to mental health, adaptation, perception, and face numerous developmental challenges and tasks related to family life, relations with peers, or education. In addition, the interest of researchers in the field of adolescence is justified for two important reasons: on the one hand, adolescence, after a relatively stable and safe period of childhood [1,2], to some extent makes a young person more vulnerable to various forms of risk, including behaviors that are incompatible with legal and social standards [3,4]. In other words, adolescence is a period in which potential crisis situations appear in biological, social, or family areas [2,5]; on the other hand, in contrast to an increased risk, adolescence brings favorable changes that contribute to a significantly better functioning young person in the cognitive sphere (e.g., the development of skills and executive functions), or in the social and mental spheres, increased self-awareness, and an increased likelihood of interactions that will shape their personality, attitudes, or behaviors [2,6]. Therefore, understanding how resilience, self-esteem, and perceived social support are interrelated can help educators and teachers to cultivate appropriate skills and competencies in young people during adolescence.

Resilience contributes to physical and mental health, as well as prevents negative effects from experienced situations and difficult events [7,8]. Resilience is a self-regulated mechanism, including cognitive, emotional, and behavioral elements. Cognitive elements are characteristic of beliefs and expectations and are related mainly to perceiving reality in terms of challenges, as well as competencies. Emotional elements of resilience related to positive affect and emotional stability. In turn, behavioral components of resilience manifest themselves as searching for new experiences and implementing various and effective strategies to cope with problems. According to this perception, psychological resilience fosters perseverance and flexible adaptation to life demands, facilitates mobilization to take remedial actions in difficult situations, and also increases tolerance for negative emotions and failures [9].

Resilience can play an important role in the functioning of children and adolescents, e.g., in terms of their ability to regulate emotions. Despite the elusive and dynamic quality of emotions, individuals have an opportunity to exercise quite a lot of control over them. In our study, the regulation of emotions refers to activities that allow an individual to monitor, evaluate, and modify the nature and course of emotional reactions [10,11]. In the emotion regulation model proposed by James Gross, it is emphasized that people can take actions to control emotions at different times, including those taken before emotions appear (antecedent-focused emotion regulation) and after the emotional reaction starts (response-focused emotion regulation) [10,12].

Although initially it was believed that positive adaptation in response to extreme life adversities is characteristic of “extraordinary” persons, research suggests that resilience is relatively common among young people [13]. In the group of young people, resilience is associated with the ability to regulate emotions and to deal with problems focused on the task and the problem itself [14,15]. In turn, difficulties in controlling emotions are negatively related to resilience [16]. In one study, it was shown that emotional dysregulation, accompanied by unfavorable childhood experiences, is greater in people with a low level of mental resistance [17]. In addition, young people characterized by a high level of resilience demonstrate a higher sense of sensibility in actions taken, a positive attitude toward life, a higher level of autonomy and confidence in each other, and more efficiency in activities required for everyday functioning. Such teenagers show greater insight and better interpersonal skills, which facilitates the establishment of warm and friendly relationships with their peers [18,19]. Moreover, young people with a higher resilience rate are more diligent, careful, helpful, and willing to cooperate [20]. It has also been shown that resilience in adolescents is positively related to self-esteem, self-assessment, and better mental health [21,22].

Research indicates that the relationship between resilience and regulation of emotions can be mediated by self-esteem [23], defined as a relatively constant disposition—that is, a conscious attitude (positive or negative) towards self [24]. High self-esteem is the conviction that one is a “sufficiently good” and valuable person, which does not necessarily mean that a person with a high self-esteem considers him- or herself to be better than others. Low self-esteem is dissatisfaction with and rejection of oneself. Self-esteem is, therefore, an evaluating and affective dimension of the image of oneself and reflects the sum of beliefs and knowledge of an individual about his or her qualities and characteristics. It is assumed that this is a coherent scheme of oneself and a detailed set of beliefs about one’s own traits of character, morals, desires, and liabilities, which eventually determine self-esteem.

According to Maslow [25], self-esteem is the basic need for a human personality. It is a source of knowledge for an individual about his or her uniqueness, which enables self-realization and self-acceptance. It is believed that self-esteem develops as a result of interacting internal factors, i.e., one’s own activities, successes, and failures, or comparisons with others; as well as external factors, i.e., the influence of other persons. Self-esteem consists of three components: cognitive, emotional, and behavioral. Cognitive components include all beliefs of an individual with respect to oneself and patterns of self-evaluation. Emotional components include love, recognition, and internal respect for an individual. The behavioral component is responsible for the level of fulfillment of one’s own needs, self-presentation, and coping with stress. Self-esteem is a key element of personality to regulate one’s own behavior and reaction to stress [26]. According to the sociometric theory, self-esteem also reflects the emotional state of an individual in his or her integration with interpersonal relationships. In other words, self-esteem is a subjective measure of the relationship between an individual and society, as well as other important persons [27]. Self-esteem is essentially associated with health, resilience, and adaptation [28].

Research has shown that high self-esteem can lead to improved ability to deal with problems and emotional regulation, while low self-esteem involves a wide range of disorders and behaviors incompatible with social norms. Young people with high self-esteem can see themselves from a positive perspective, are more prone to effective emotional regulations, and have increased confidence and optimism [29]. Kapikiran and Acun-Kapikiran found self-esteem mediated resilience and depression symptoms among young people [30]. Moatsou and Koutra found that self-esteem fully mediates the relationship between resilience and suppression of expression. In this trend, as well as in relation to the review of the literature, it seems that self-esteem can mediate resilience and emotional regulation [31].

Perceived social support is important in adolescence as a factor that protects young people. Social support is described as the support an individual receives through social networks with other people [32]. In our study, social support covers areas such as the sense of being helped and social networks, as well as emotional support; that is, the feeling of being understood and accepted. Generally, social support has a beneficial effect on relieving individual psychological pressure, regulating negative emotions, providing positive emotional reinforcement, and promoting mental health [33]. Social support can also be considered one of the most important external resources buffering the negative effects of stressors, and some studies emphasize a positive impact of social support on self-esteem. Recent studies indicate that the relationship between resilience, self-esteem, and emotional regulation may be additionally mediated by the perception of social support. Social support has been extensively studied from various theoretical perspectives [34]. In this approach, the main effect model and the stress-buffering model are recognized as two important theoretical frameworks that include social support in the psychosocial functioning of an individual, which derives from the theory of stress and coping [35]. The main effect model shows that social support has a direct impact on adaptation and health results regardless of the level of stress [36]. The stress-buffering model indicates that social support plays the role of a buffer or protector (mediator or moderator), facilitating the improvement of physical and mental health [37]. In this study, the stress-buffering model is used to demonstrate the mediating effect of social support on the relationship between resilience, self-esteem, and emotional regulation.

Effective emotional regulation and resilience appear to be important variables in the stress and experiences of young people during adolescence [38,39]. The aim of this study was to test a mediation model that takes into account self-esteem and perceived social support as mediators of the relationship between resilience and emotional regulation. In line with the review of the literature set forth above, we decided to test the hypothesis that the relationship between resilience and emotional regulation is serially mediated by self-esteem (M1) and perception of social support (M2) in adolescents. 

## 2. Materials and Methods

### 2.1. Participants and Procedure

The study involved 251 adolescents aged between 14 and 19 years (M = 16.85, SD = 1.09), with 61% being girls. The study was approved by the Ethics Committee of the Faculty of Educational Sciences, the University of Białystok (# 001/2021). The data were collected in June 2021 from a group of students from three school dormitories in Poland: the Dormitory of the Schools of Commerce and Economics in Białystok, the Dormitory of the Secondary School No. 10 in Białystok, and the Dormitory of the Complex of Agricultural Schools of the Practical Training Centre in Białystok. The study procedure consisted of completing paper-and-pencil questionnaires to measure resilience, self-esteem, and to answer questions about perceived social support and emotional regulation. Participation in the study was anonymous and voluntary. Consent was required of the adolescents and their legal guardians. The rules of the sanitary regime were followed during the data collection stage in connection with the ongoing COVID-19 pandemic. All collected questionnaires were determined to be complete and were further analyzed.

### 2.2. Measurement

The Resilience Scale (RS) by Ogińska-Bulik and Juczyński [9], which consists of 18 items, was used as an adaptive mechanism to assess resilience. The respondents, using a 5-point Likert-type scale (from definitely no to definitely yes), indicated to what extent they agreed with individual statements. The internal consistency of the scale is satisfying, with Cronbach α = 0.82. Sample items: “I am open to new experiences”, “I consider myself a strong person”.

The Rosenberg Self-Esteem Scale (SES) was used to measure self-esteem [40] as a relatively constant disposition understood in terms of a conscious attitude (positive or negative) towards oneself. The scale consists of 10 diagnostic questions. The respondent used a 4-point scale to indicate to what extent he or she agreed with each statement. The internal consistency of the scale is high, with Cronbach α = 0.81 to 0.83. Sample items: “I believe that I have many positive traits”, “Overall, I am pleased with myself”.

An original scale applied by the authors, which consisted of three items, was used to measure perceived social support. Exploratory factor analysis revealed a univariate structure of the questionnaire. The single-factor model of this scale explained 52% of the variance. The created factor was called “perceived social support” and demonstrated a satisfactory internal coherence for the scale (Cronbach α = 0.62; McDonald Ω = 0.61). Sample items: “When I have a problem I turn to my family members”, “When I have a problem I turn to my friends“.

The assessment of the ability to regulate emotions was measured using the original three-item scale applied by the authors. The factor analysis revealed a univariate structure for the statements. The single-factor model of this scale explained 59% of the variance. The created factor was called “emotional regulation” and demonstrated a satisfactory internal coherence for the scale (Cronbach α = 0.66; McDonald Ω = 0.71). Sample items: “I know and use constructive ways to deal with anger”, “When I have a problem, I can control my emotions”.

### 2.3. Statistical Analyses

The analyses were carried out with the use of the IBM SPSS Statistics 27 software and the PROCESS plug-in version 4.0 for the analysis of mediation effects. The evaluation of the relationships between the variables was performed using the r-Pearson (r) correlation analysis. Per Cohen, the absolute value of a correlation is equivalent to its effect size, with those under 0.10 being trivial, those between 0.10−0.30 having a small/weak effect, those between 0.30−0.50 having a medium effect, and those >0.50 having large effect [41]. The analysis of the effect of serial mediation (Model 6) was performed using the bootstrap method. The bootstrap analysis sample size was 5000, and the mediation effect test is significant when it does not contain zero under the 95% confidence interval (CI). The significance level was set at *p* ≤ 0.05.

## 3. Results

The mean values obtained in the study for resilience, self-esteem, perceived social support, and emotional regulation, together with standard deviations, threshold values, and correlations, are shown in Table 1. Correlation analysis showed statistically significant relationships between self-esteem and resilience; between perceived social support and resilience and self-esteem; and between emotional regulation and resilience, self-esteem, perceived social support, and emotional regulation. In this study, age correlated positively with resilience (r = 0.17, *p* = 0.008). Sex (0 = female; 1 = male) was correlated with resilience (r = 0.19, *p* = 0.002), self-esteem (r = 0.17, *p* = 0.008), and emotional regulation (r = 0.18, *p* = 0.004). None of the other sociodemographic factors affected the results in a statistically significant way.

Bootstrap sampling analysis showed statistically significant serial mediation. The model assessed the link between resilience (X), self-esteem (M1), perceived social support (M2), and emotional regulation (Y). As a result of the analysis, a positive direct relationship between resilience and emotional regulation was observed (total effect; B = 0.061; SE = 0.012; 95% CI = 0.037, 0.086; r2 = 0.09). After including mediators of self-esteem and perceived social support in the analysis, the relationship coefficient decreased but was still statistically significant (direct effect; B = 0.046; SE = 0.016; 95% CI = 0.015, 0.078; r2 = 0.12). Resilience also proved to be a positive predictor of self-esteem (B = 0.346; SE = 0.025; 95% CI = 0.296, 0.396; r2 = 0.43) and perceived social support (B = 0.038; SE = 0.012; 95% CI = 0.014, 0.063; r2 = 0.04).

The analyses revealed that the indirect effect of resilience on emotional regulation through self-esteem is not significant (B = 0.045; SE = 0.062; 95% CI = −0.076, 0.166). The indirect effect of resilience on emotional regulation via perceived social support was also found to be not significant (B = −0.002; SE = 0.015; 95% CI = −0.032, 0.029).

Finally, the study assessed the indirect impact of resilience on emotional regulation through self-esteem and perceived social support. The relationship was significant with a point estimate of 0.030 (testing serial multiple mediation; SE = 0.016, 95% CI = 0.004, 0.068). A visualization of the mediation model is presented in Figure 1. The alternative model, where the mediator order has been reversed (M1: perceived social support; M2: self-esteem), was not significant, with a point estimate of 0.003 (SE = 0.004, 95% CI = −0.005, 0.010).

## 4. Discussion

The aim of this study was to evaluate the relationships between resilience, self-esteem, perceived social support, and emotional regulation. Our results suggest that self-esteem and perceptions of social support may mediate the relationship between resilience and emotional regulation. The obtained data are consistent with previous reports, which indicate positive relationships between these variables in adolescents [23,42]. According to researchers, the regulation of emotions refers to activities that allow an individual to monitor, evaluate, and modify the nature and course of emotional response; resilience, self-esteem, and perceived social support determine its growth [16,17]. Moreover, our data suggest that resilience may increase the level of emotional regulation skills in adolescents, which also supports previous empirical findings [39]. 

We have shown that the relationship between resilience and emotional regulation can be serially mediated by perceived social support and self-esteem (partial mediation). According to the data obtained, resilience can increase the ability to regulate emotions by enhancing self-esteem and perception of social support. The alternative model, with an inverted order of mediators, turned out to be irrelevant, which also suggests that self-esteem should be viewed as a relatively constant attitude towards oneself that may lead to an increase in subjective perceptions of social support. Moreover, it should be noted that in our study, only the serial mediation model turned out to be significant. However, we did not show significant effects of self-esteem and perceived social support in single models of mediation between resilience and emotional regulation. Therefore, it seems that these two variables—individual and environmental—are equally important and necessary for proper adaptation to life experiences and emotional regulation, which is also confirmed by the findings of Kapikiran and Acun-Kapikiran [30].

In the study, it turned out that self-esteem was positively related to resilience, perceived social support, and regulation of emotions but also played a significant intermediary role in relationships between the analyzed constructs. Although in numerous studies it was checked whether self-esteem mediates the impact on other psychological variables, none of the previous analyses considered the determination of the role of self-esteem in the relationship between resilience and regulation of emotions. As expected, it turns out that self-esteem is a serial mediator in the analyzed relationship, which means that it is a critical construct in the psychosocial development of adolescents. To be more specific, the ability to regulate emotions was directly related to a higher resilience level, and this relationship was also partially mediated by self-esteem and perceived social support. The research results suggest that people who control their emotional state are considered more valuable and, as a result, more often perceive social support and can deal more effectively with traumatic events. 

It should be noted that adolescence is a period in life that is characterized by biological, behavioral, and psychological changes taking place under the influence of social or family contexts [1,2]. Along with the change and expansion of social relationships in the family, school, and peer areas, interdependence and influence among individuals and various social groups increase. In this respect, perceived social support is closely related to the emotional and social development of teenagers, which is confirmed in our findings [43,44]. Thus, perceived social support can influence the regulation of emotions by young people and improve their problem-solving skills, promoting adaptation behaviors as a result. 

In this study, age was positively correlated with resilience, which is in line with earlier findings [45,46]. However, sex was correlated with resilience, self-esteem, and emotional regulation. The results of previous studies also indicate that sex has an important role in the context of the analyzed variables [47,48,49]. 

In general, these results are consistent with the results of other research and suggest that the inclusion of modules reinforcing resilience, self-assessment, and perception of social support in prophylactic and therapeutic programs may be beneficial to improving the effects [50,51]. In the face of our own research and data acquired from the literature, it seems that psychoeducational interventions that strengthen self-esteem and provide knowledge about resources required to deal with everyday stressors build strong self-esteem and allow young people to prosper during adolescence despite difficult and demanding situations. In this way, they turn out to be beneficial for adolescents because they promote and strengthen the ability to regulate emotions. Our research emphasizes the role of diagnosis in a positive sense, which focuses not on weaknesses but on strengths and positive features such as resilience and self-esteem. It is worth highlighting that our findings are but a step that encourages continued research in the field of educational psychology, and especially to characterize myriad variables that can be used for the personal and educational development of adolescent students. 

Despite its strengths, our research also has some limitations. First of all, the cross-sectional project prevents the unambiguous determination of cause-and-effect relationships. Secondly, the self-reporting nature of the study makes the declared intensification of variables slightly different from their actual level. Thirdly, the inclusion of a larger number of sociodemographic data (e.g., school, housing conditions) would provide a fuller image as far as determinants of individual phenomena are concerned. Despite these limitations, our study provides new data in the field of resilience, self-esteem, perceived social support, and regulation of emotions. Future research should be longitudinal, integrate self-reported and interpretive measures, and include a gender-balanced sample and more representative adolescent samples for a better understanding of the relationship between emotional regulation, perceived social support, self-esteem, and resilience.

## 5. Conclusions

In this study, the relationship between resilience, self-esteem, perceived social support, and regulation of emotions was assessed using a serial mediation model. Our data indicate that resilience, self-esteem, and perceived social support can be functional in regulating effect. Resilience increases the ability to regulate emotions because it enhances self-esteem and the perception of social support. Those two variables seemed as relevant as resilience to achieve emotional regulation, preparing the adolescent to face life experiences. The findings have an applicable value. They can be used to develop preventive and educational programs, as well as therapeutic interventions. The obtained results show that interventions aimed at resilience can improve self-assessment and perceived social support and thus favor a high level of emotional regulation skills in the adolescents. 

## Figures and Tables

**Figure 1 ijerph-19-08007-f001:**
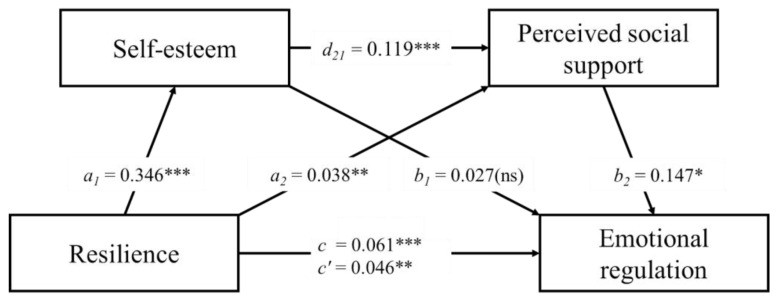
Results of the serial multiple mediational model, *** *p* < 0.001; ** *p* < 0.01; * *p* < 0.05; ns = not significant. Values shown are unstandardized coefficients.

**Table 1 ijerph-19-08007-t001:** Mean and correlations of test results (N = 251).

	Min	Max	M	SD	1.	2.	3.
1. Resilience	21	72	49.67	10.08	1		
2. Self-esteem	10	40	27.8	5.34	0.65 ***	1	
3. Perceived social support	0	9	5.41	2.01	0.19 **	0.31 ***	1
4. Emotional regulation	0	9	5.39	2.04	0.30 ***	0.26 ***	0.21 ***

*** *p* < 0.001; ** *p* < 0.01.

## Data Availability

The data presented in this study are available on request from the corresponding author agreed to the published version of the manuscript.

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
