# Peer review of "Resilience and Regulation of Emotions in Adolescents: Serial Mediation Analysis through Self-Esteem and the Perceived Social Support"

_ijerph, 2022, doi:10.3390/ijerph19138007_

Round 1

Reviewer 1 Report

It is an excellent paper. Before publication, I have only minor suggestions:

Please provide sample statements for measures and full items of your scale.

Please describe the results of the exploratory factor analysis for your scale in the Methods section.

In the Results section is the sentence "Sex was correlated with resilience (r = 0.19, p = 0.002),". It would be clear if you provide coding technique in brackets, e.g. 0 = male, 1 = female.

Author Response

We implemented all the suggestions of the reviewers.

Reviewer 2 Report

The manuscript is concise and well written. The research was well designed and conducted, although it does not present a novelty. The work will contribute to reinforcing the importance of the constructs and the correlations found in this research. 

The abstract should mention main correlation indexes plus statistical significance, to highlight the magnitude of the findings, calling attention to the contribution of this research.

Besides,  change final phrases stressing that resilience increased emotions regulation ability because it enhances self-esteem and the perception of social support, but these last  two variables seemed as relevant as resilience to achieve emotional regulation, preparing the adolescent to face life experiences.

Author Response

(The authors gave the same response as above.)
